# DPPA as a Potential Cell Membrane Component Responsible for Binding Amyloidogenic Protein Human Cystatin C

**DOI:** 10.3390/molecules29153446

**Published:** 2024-07-23

**Authors:** Igor Zhukov, Emilia Sikorska, Marta Orlikowska, Magdalena Górniewicz-Lorens, Mariusz Kepczynski, Przemyslaw Jurczak

**Affiliations:** 1Laboratory of Biological NMR, Institute of Biochemistry and Biophysics, Polish Academy of Sciences, 02-106 Warsaw, Poland; igor@ibb.waw.pl; 2Department of Organic Chemistry, Faculty of Chemistry, University of Gdansk, 80-308 Gdansk, Poland; emilia.sikorska@ug.edu.pl; 3Department of Biomedical Chemistry, Faculty of Chemistry, University of Gdansk, 80-308 Gdansk, Poland; marta.orlikowska@ug.edu.pl; 4Faculty of Chemistry, Jagiellonian University, 30-387 Krakow, Poland; magdalena.gorniewicz@doctoral.uj.edu.pl (M.G.-L.); kepczyns@chemia.uj.edu.pl (M.K.); 5Doctoral School of Exact and Natural Sciences, Jagiellonian University, Profesora Stanisława Łojasiewicza 11, 30-348 Krakow, Poland; 6Laboratory of Mass Spectrometry, Intercollegiate Faculty of Biotechnology UG&MUG, University of Gdansk, 80-307 Gdansk, Poland; 7Biomacromolecule Research Team, RIKEN Center for Sustainable Resource Science, Wako-shi 351-0198, Saitama, Japan

**Keywords:** human cystatin C, DPPA, liposome, NMR, molecular dynamics, differential scanning calorimetry, interactions

## Abstract

A phospholipid bilayer is a typical structure that serves crucial functions in various cells and organelles. However, it is not unusual for it to take part in pathological processes. The cell membrane may be a binding target for amyloid-forming proteins, becoming a factor modulating the oligomerization process leading to amyloid deposition—a hallmark of amyloidogenic diseases—e.g., Alzheimer’s disease. The information on the mechanisms governing the oligomerization influenced by the protein–membrane interactions is scarce. Therefore, our study aims to describe the interactions between DPPA, a cell membrane mimetic, and amyloidogenic protein human cystatin C. Circular dichroism spectroscopy and differential scanning calorimetry were used to monitor (i) the secondary structure of the human cystatin C and (ii) the phase transition temperature of the DPPA, during the protein–membrane interactions. NMR techniques were used to determine the protein fragments responsible for the interactions, and molecular dynamics simulations were applied to provide a molecular structure representing the interaction. The obtained data indicate that the protein interacts with DPPA, submerging itself into the bilayer via the AS region. Additionally, the interaction increases the content of α-helix within the protein’s secondary structure and stabilizes the whole molecule against denaturation.

## 1. Introduction

A phospholipid bilayer serves as a building block of the outer wall of most cells and organelles in living organisms. Even though rather inconspicuous, it plays a significant role as a barrier separating the cell from factors present in the outer environment. The bilayer plays multiple essential roles in an organism. Depending on the location, its lipid composition differs based on the necessary functions and properties [1]. In some specialized cases, lipids can even be organelle- or location-specific. Such a case can be observed for cardiolipin in mitochondria [2], lysobisphosphatidic acid in late endosomes [3], or phosphatidylserine found almost exclusively in the cytoplasmic leaflet of the human plasma membrane [4,5]. Besides the functions the phospholipid bilayer contributes based on its lipid composition, it also serves an essential role of a host, providing a proper environment for proteins and carbohydrates responsible, e.g., transmembrane trafficking or signal transduction [6].

Despite its importance for the proper performance of an organism, the phospholipid membrane is also involved in pathological processes leading to harm and disease by playing an essential role in the mechanisms leading to the oligomerization of amyloid-forming proteins [7]. An amyloid is an aggregated insoluble form of a protein or peptide that accumulates in different parts of an organism, causing severe damage. Currently, c.a. forty amyloid proteins have been discovered and described. The correlation between the presence of cell membranes and their oligomerization was proved in quite a few cases [8,9,10]. As a result, the bilayer is partially involved in the processes that eventually lead to diseases such as Alzheimer’s or Parkinson’s disease. The influence of model liposomal cell membrane mimetics on the oligomerization of hCC was also studied [11,12]. Even though no acceleration of the process was observed, the protein–mimetic complexes were formed. Currently, two possible mechanisms of oligomerization of amyloid-forming proteins involving cell membranes have been proposed [7]. They both assume that an annular oligomer is formed due to a protein–membrane interaction but differ regarding the level of participation of the membrane in the process. The main feature of annular oligomers is their ability to interact with biological membranes and disturb their integrity by forming channels. As a result, they are indicated as a potential cause of toxic properties exhibited by amyloidogenic proteins [7]. The assembly into annular oligomers was observed for, i.a., the following amyloidogenic proteins: amyloid β peptide [13], immunoglobulin light chain [14], and human cystatin C [15,16].

Human cystatin C (hCC) is a protein common for all body fluids [17]. It is physiologically relevant due to its regulatory activity toward inter- and intramolecular cysteine proteases [18] and partial role in neuroprotective mechanisms [19,20,21]. However, under certain conditions, it may be prone to dimerization and oligomerization [22,23,24]. Its highly amyloidogenic Leu^68^→Gln mutant is correlated with a dominant hereditary disorder called hereditary cystatin C amyloid angiopathy [25]. Brain strokes and death of patients at a young age characterize this disease.

Considering available data, the interactions between human cystatin C and biological membranes in the context of the mechanisms driving the amyloid formation are of biological relevance. However, experimental research involving membranes is somewhat complex and nontrivial due to biological membranes’ complex and multicomponent structure. Therefore, natural membranes are often substituted to facilitate experimental measurements with model membrane mimetic structures in the form of phospholipid bilayer structures [26,27]. It has been shown that hCC interacts with DMPC and POPG phospholipids [11,12]. Following these data, in the present study, we have monitored the influence of 1,2-dipalmitoyl-sn-glycero-3-phosphate (DPPA) bilayer on the structure of V57G human cystatin C mutant (hCC V57G). DPPA was selected for the study as it is a significant component of cell membranes [28]. This anionic lipid with a phosphate headgroup serves essential biophysical, metabolic, and signaling roles, including a critical role in signal transduction and pH sensing [29,30]. Importantly phosphatidic acid (PA), a headgroup component of DPPA, has also been implicated in many aspects of animal cell biochemistry and physiology, including cell proliferation and differentiation, cell transformation, tumor progression, and survival signaling [31].

The main focus of this project was to determine if the DPPA bilayer exhibits affinity toward the hCC protein. This would indicate that the DPPA phospholipid is a potentially crucial cellular component involved in protein–membrane binding. Consequently, it is a potential factor facilitating the protein’s homeostatic transmembrane trafficking or pathological oligomerization modulated by the membrane. During the study, we monitored the structural changes in human cystatin C during interactions between the protein and the DPPA cell membrane mimetic, with circular dichroism (CD) spectroscopy. The differential scanning calorimetry (DSC) allowed the detection of changes in the phase transition temperature of the DPPA phospholipid bilayer caused by interactions with the hCC V57G protein. Further, the nuclear magnetic resonance (NMR) spectroscopy allowed us to determine which parts of the protein are responsible for interacting with the DPPA bilayer. Finally, the molecular dynamics (MD) simulations provided a theoretical structure representing hCC V57G–DPPA interaction and supported the data obtained from NMR experiments.

## 2. Results and Discussion

For better clarity, this section briefly explains the reasoning behind the selection of the hCC V47G mutant instead of the wild-type protein. The previous extensive studies on the human cystatin C protein variants [18,32,33,34,35] show that all the hCC mutants exhibit properties similar to the wild-type protein. The proteins (including hCC V57G) retain their original activity but differ in the propensity for oligomerization. This is especially important in the case of NMR experiments. To date, the NMR structure was determined only for the hCC V57G monomeric mutant [35]. This is due to the difficulties caused by a partial dimerization of the wild-type protein during the NMR experiments and close to identical properties of the subunits within the structure of the protein dimer. As a result, the signals representing specific amino acid residues within the subunits overlap, and the signal assignment is difficult. The fast development of NMR techniques allowed us to start work on the NMR structure of the stable dimer of human cystatin C—hCC V57P (unpublished data). The wild-type protein is next in turn. Nevertheless, when the experiments described below were performed, the NMR assignment of amino acid residues was available only for the hCC V57G structure. Therefore, using NMR techniques for studies on interactions between phospholipid membranes and the hCC WT was impossible. Further, the molecular dynamics simulations are not yet powerful enough to simulate the dimerization of a protein, so the studies on oligomerization of hCC would not, in this case, be relevant. The mutations influencing the oligomerization propensity of hCC also do not seem to affect the interaction (data unpublished) as they occur within the L1 loop [33,34,35], and the lipid membranes interact with the AS loop [11,12], located on the other side of the protein. Therefore, it seems that the NMR spectroscopy and molecular dynamics simulations are only relevant in the case of the V57G mutant. Finally, we believe that adding the data on wild-type protein or other hCC mutants to the circular spectroscopy or differential scanning calorimetry sections would only introduce unnecessary chaos to the manuscript and could confuse a reader.

### 2.1. Circular Dichroism

Circular dichroism (CD) spectroscopy experiments were performed to visualize the changes in the hCC V57G protein’s secondary structure resulting from the protein’s interactions with the phospholipid bilayer of DPPA liposomes. The interactions with the DPPA caused an increase in the strength of the signals in CD spectra at wavelengths of ca. 209 nm and 221 nm (Figure 1), indicating an increase in the content of the α-helix in the protein structure. The formation of an α-helix and an increase in order within the protein structure was most probably induced by the interaction of the AS region (marked in Appendix A), a relatively unorganized fragment of the protein, with the hydrophobic environment of the DPPA bilayer interior. This theory was later confirmed with the NMR data, which shows that the protein submerges itself into the DPPA lipid bilayer with the AS region. Additionally, the later-obtained dynamic scanning calorimetry data support the formation of the α-helix within the protein structure, showing a significant decrease in the protein’s entropy change value during the interaction with the bilayer. When compared to our previous research on hCC V57G–POPG and hCC V57G–DMPC interactions [11,12] this is the first time phospholipid liposomes have caused changes in the secondary structure of the hCC protein.

### 2.2. Differential Scanning Calorimetry

The effect of the hCC V57G protein on the thermotropic behavior of the DPPA phospholipid bilayer was monitored with differential scanning calorimetry (DSC). The tests were carried out for liposomes consisting of the DPPA phospholipid with and without hCC V57G and for a pure protein solution. The DSC curve for pure DPPA was characterized by the presence of one endothermic peak located at 59.8 °C (Figure 2), which is associated with the main phase transition of the lipid bilayer—the transition from the gel phase (L_β′_) to the liquid crystal phase (L_α_) [36]. The DSC thermogram for the pure hCC V57G solution showed a single endothermic peak centered at 82.6 °C (Figure 2), which represents the transition of the protein from the folded to the unfolded state—denaturation (T_d_) [37]. The DSC curve registered for the solution containing DPPA liposomes and hCC V57G protein contained two peaks, one corresponding to the phase transition in the liposomes and the other in the protein (Figure 2). The presence of hCC V57G had a significant impact on the thermotropic behavior of the DPPA bilayer, indicating protein–DPPA interactions. This correlates with a general tendency of melting profiles of lipid membranes to change upon protein binding [38]. The data show a shift in the DPPA phospholipid bilayer melting temperature (T_m_) toward higher values (Table 1), indicating purely electrostatic interactions between the positively charged protein and the negatively charged bilayer. Increased T_m_ also shows the disruption of lipid–lipid interactions and the formation of a protein–lipid complex [39]. The increase in the T_m_ of the DPPA membrane was accompanied by a decrease in the enthalpy change (ΔH) (Table 1). This phenomenon occurs probably due to the weakening of interactions between acyl chains within the DPPA bilayer after the hCC V57G protein adsorption [40].

The peak width (ΔT_1/2_) and peak shift are measures of the cooperativity of the phase transition [41]. The addition of hCC V57G protein to the DPPA liposome solution resulted in a decrease in ΔT_1/2_ (narrowing of the peak) and a shift in the peak maximum toward a higher temperature (62.1 °C) (Table 1). This can be explained by assuming that the protein stabilized the lipid bilayer structure by forming stable protein–lipid complexes, which increased the homogeneity and coherence of the bilayer structure [42], and increased the number of lipid molecules simultaneously participating in the phase transition (increased cooperativity) [43]. Stabilization of the structure of the lipid bilayer is directly correlated with the proper functioning of biological membranes, especially in the case of transport and transmitter functions [44]. In this case, the protein regulated interactions between lipids through electrostatic or hydrophobic interactions, van der Waals forces, and hydrogen bonding, unifying the structure of the lipid bilayer [45]. The increase in uniformity and coherence of the structure of the phospholipid bilayer in a complex with the hCC V57G protein was confirmed by the entropy change (ΔS). The ΔS of the protein–lipid system is smaller than the pure lipid bilayer. This is probably caused by increased order within the structure due to a more parallel organization of lipid chains after interaction with the protein. The decreased ΔS value for the protein also agrees with CD data, which shows an increase in order in the protein structure in the presence of the DPPA bilayer. The analysis of thermal profiles and thermodynamic parameters registered for the peak corresponding to the hCC V57G protein further confirms that the protein–lipid interactions stabilize the protein structure. This is reflected in a slight decrease in the T_d_ of the protein in the presence of DPPA phospholipids, which, in turn, was most probably caused by the migration of the protein into the hydrophobic interior of the lipid bilayer (in agreement with NMR data shown below) and the formation of structures surrounding the protein and protecting it against denaturation. The hCC V57G thermogram shows an apparent blurring and lowering of the protein denaturation peak in the presence of the DPPA phospholipids (Figure 2). This indicates the stabilization of the protein structure due to the formation of a protein–lipid complex.

### 2.3. Molecular Dynamic Processes Occurring in the hCC V57G Protein in the Presence of DPPA Phospholipid Bilayer

The relaxation data were collected for backbone resonances assigned to 81 amide groups of the hCC V57G protein. Similar to our previous studies [11], the rotational diffusion tensor can be described as an axially symmetrical model of the protein since D_∥_ = 2.77 ± 0.03 × 10^−7^ cm^2^/s and D_⊥_ = 1.63 ± 0.02 × 10^−7^ cm^2^/s. The overall tumbling time, calculated as τ_R_ = 1/(2D_∥_ + 4D_⊥_), equaled τ_R_ = 8.29 ± 0.07 ns. When compared to the data obtained for the hCC V57G (7.45 ± 0.11 ns) [35], or hCC V57G interacting with the DMPC-d_56_ lipid bilayer (7.41 ± 0.06 ns) [12], the τ_R_ value was slightly increased. At the same time, it was similar to the data obtained for the hCC V57G interacting with the POPG lipid bilayer (8.35 ± 0.07 ns) [11]. The increased τ_R_ value suggests that interactions between hCC V57G and the bilayer of DPPA or POPG lipid liposomes are more robust than those observed between hCC V57G and DMPC phospholipid liposomes.

### 2.4. Analysis of ^15^N Relaxation Data with Spectral Density Mapping Approach

The relaxation rates and steady-state ^1^H-^15^N *NOE* data were collected for the hCC V57G protein at the magnetic field of 11.7 T (Appendix A). The ^15^N *R*_1_ relaxation rates of amino acid residues in the hCC V57G protein sequence are uniform, with mean values of 1.766 ± 0.038 s^−1^. Smaller *R*_1_ values were observed only in the flexible L1 and L2 structural loops (Gly59, Val104, Thr109) and other residues involving Asp40, Gly69, and Ala120. An increase in relaxation rates evaluated, based on the ^15^N *R*_2_ values, was observed for the His43, Ser44, Asp81, and Asp119. At the same time, remarkably decreased *R*_2_ values were observed for the Gly59, Gly69, and C-terminal Ala120.

The spectral density mapping approach was used to analyze the obtained ^15^N relaxation data [46]. The hCC V57G–DPPA interactions were represented by increased molecular dynamics processes within a low frequency (ms–μs) time frame. According to our analysis, only two amino acid residues undergo such processes—Asp81 and Asp119 (Figure 3). They are also both involved in additional dynamic processes resulting from conformational exchange motions.

### 2.5. Determination of Amino Acid Residues Buried in the DPPA Phospholipid Bilayer—Paramagnetic Relaxation Enhancement Approach

The amino acid residues of the hCC V57G protein buried into the DPPA phospholipid bilayer were determined based on the effect of paramagnetic relaxation enhancement (PRE) [11,47]. The addition of Cr(AcAc)_3_ paramagnetic compound to the ^15^N labeled protein sample resulted in a substantial increase in resonance linewidth of the signals representing the solvent-exposed amino acid residues in the ^1^H-^15^N HSQC spectrum (Appendix A). The amino acid residues involved in the interaction with the DPPA phospholipid bilayer were derived from the normalized amplitude changes extracted from ^1^H-^15^N HSQC data (Figure 4A). They were selected based on the amplitude change parameter, which, in this case, was lower than 10%. Based on the hCC V57G structure analysis, we concluded that most of those amino acid residues could be found in the AS region of the hCC V57G protein (Figure 4B).

The obtained data indicate that Asp81 and Asp119 may form a pair, which is the main force driving the hCC V57G–DPPA interactions. But other residues (including Arg45, Thr72, Cys73, Gln77, Asn79, His86, Lys94, and Ala96) (Figure 4A) are also involved in the interaction. All the above-mentioned amino acid residues taking part in interaction with the hCC V57G protein were marked in Figure 4B. The obtained data agree with the CD spectroscopy results, which indicate the potential involvement of the AS region in the interaction. Further, the location of the amino acid residues interacting with the lipid bilayer suggests that the protein may be attached to or submerged into the membrane with the AS region, which agrees with DSC data, and moves relatively easily in the extracellular space. As a result, the protein may lie down on the bilayer surface, which would explain the involvement of the amino acid residues outside of the AS region in the interaction. This theory was later supported with molecular dynamics simulation data.

### 2.6. Coarse-Grained Molecular Dynamics Simulations

Coarse-grained molecular dynamics simulations (CG MD) were employed to predict the interactions between the hCC V57G protein and the DPPA lipid bilayer. Two different approaches were used to investigate possible ways of protein–bilayer binding. In the first approach (Figure 5), the protein was embedded in the DPPA phospholipid bilayer (models I–II), which made it possible to follow the mutual adaptation of the protein and the membrane during the CG MD simulations. In the second approach (Figure 6), the protein was placed above the outer leaflet of the pre-equilibrated DPPA lipid bilayer in different initial orientations and distances relative to the membrane (models III–VI), which allowed monitoring of the spontaneous adsorption of the protein to the membrane.

In models I and II, the protein quickly emerged from the membrane core during the first steps of the CG MD simulations (up to 50 ns) and localized itself on the membrane–water interface in a different final orientation. In model I, the β-strands of the hCC V57G molecule were facing the membrane surface, while the 21–35 α-helical fragment (α1) was located above the β-strands. Therefore, it was unable to interact with the DPPA lipids. A detailed analysis of the distance between individual amino acids and DPPA lipids showed that the C-terminus, the AS region, and fragments of the β-strands located in the AS pole of the protein were involved in its anchoring to the DPPA bilayer (Figure 7). In model II, the protein was arranged in the opposite way, with the 21–35 α-helix (α1) involved in the binding to the DPPA lipid bilayer. In addition, the N-terminus, the L1 and L2 loops, and the fragments of the β-strands connected to these loops anchored the protein in the membrane. It is worth noting that in model II, the protein level immersed in the lipid bilayer was the greatest among all the simulations. This led to a significant reduction in membrane thickness and an increase in the area per lipid (APL) directly at the binding site (Appendix A). Moreover, a local mean curvature analysis showed an overall bending of the bilayer directly beneath the protein.

The simulations of the spontaneous adsorption of the hCC V57G protein on the DPPA lipid bilayer (models III–VI) also revealed two ways of protein binding. Model IV was more or less consistent with model I, whereas models III, V, and VI were consistent with model II. Some emerging differences between the corresponding models resulted from the final tilt of the protein with respect to the membrane. Thus, the analysis of the tilt angles of both α-helical fragments, 21–35 (α1) and 81–90 (α2), with respect to the membrane normal (z-axis) (Figure 8) revealed a high convergence in models I, III, V, and VI, while a comparison of models I and IV showed a similar arrangement of the 81–90 α-helix (α2), but a significantly different arrangement of the 21–35 α-helix (α1). The discrepancies in models I and IV resulted from various mutual arrangements of both helical fragments (Figure 8C), which seem to be a direct consequence of the AS region flexibility (Figure 8D). In the case of model I, the establishment of interactions between the hCC V57G and the DPPA bilayer during CG MD simulations probably induced a deflection of the protein backbone in the AS region, and thus the 81–90 α-helix (α2) itself. Model I is also the only one in which the Asp81 and Asp119 amino acid residues, indicated in the NMR study as a potential motif driving the interactions between hCC V57G and the DPPA lipid bilayer, were located relatively close to the bilayer surface. In other cases, only one of the residues is directed toward the membrane, while the other is exposed to the aqueous environment. This is probably due to anchoring the protein with the AS region to the bilayer surface and its relative freedom of movement in extracellular space (supported by NMR data). The NMR data provide a structure that presents an average of multiple states of the protein in solution. This averaged structure indicates the Asp81 and Asp119 pair as the driving force of the interaction. However, due to the movement allowing the protein to assume different positions in space, the Asp81/Asp119 motif changes its position in relation to the bilayer. Therefore, it is possible that in some positions both amino acid residues in the motif are submerged into the bilayer. In others, only one of the residues interacts with the membrane, and the other is exposed to the extracellular space. In none of the CG MD simulations did the protein binding induce any significant changes in the lateral diffusion coefficients of the DPPA lipids, the values of which ranged from 6.48 ± 0.10 × 10^−7^ to 7.00 ± 0.04 × 10^−7^ cm^2^/s in the systems with the protein, compared to 6.44 ± 0.01 × 10^−7^ cm^2^/s for the control simulation without the protein.

### 2.7. Summary

Obtained results show that during the interaction with the DPPA bilayer, an α-helix is formed, and the order within the protein structure increases. These results are confirmed by the increased strength of the signals in CD spectra at wavelengths of ca. 209 nm and 221 nm. The decreased ΔS value for the protein observed in the DSC experiment agrees with CD data. It shows increased order in the protein structure in the presence of the DPPA bilayer. Further analysis of thermal profiles and thermodynamic parameters registered for the peak corresponding to the hCC V57G protein confirms that the protein–lipid interactions stabilize the protein structure. This is reflected in a slight decrease in the T_d_ of the protein in the presence of DPPA phospholipids. The formation of an α-helix and an increase in order within the protein structure was most probably induced by the interaction of the AS region, a relatively unorganized protein fragment, with the hydrophobic environment of the DPPA bilayer interior. This theory was confirmed by the NMR data, which shows that the protein submerges itself into the DPPA lipid bilayer with the AS region. Further, the MD simulations show that Asp81 and Asp119 amino acid residues, indicated by the NMR data as a potential motif driving the interactions between hCC V57G and the DPPA lipid bilayer, were located relatively close to the bilayer surface. This probably results from anchoring the protein with the AS region to the bilayer surface.

## 3. Materials and Methods

### 3.1. Protein Expression

The DNA of the hCC V57G protein was obtained via site-directed mutagenesis. The native hCC V57G protein was expressed using standard LB medium and temperature-induced expression, according to the protocol described earlier [18]. The single ^15^N-labeled hCC V57G protein was expressed as described earlier with the use of single labeled minimal media containing ^15^NH_4_Cl [35]. The isolation and purification of both labeled and unlabeled proteins was performed as described earlier with the use of ion exchange chromatography and size exclusion chromatography [18].

### 3.2. Circular Dichroism Measurements

The lipid sample was formulated as described earlier [12]. The DPPA powder was mixed with a PBS solution and exposed to fifteen freeze–thaw cycles. The Litesizer 500 particle analyzer (Anton Paar GmbH, Austria) was used to determine the size of the obtained phospholipid structures (Appendix A). The hCC V57G was dissolved in PBS buffer and mixed with DPPA liposome solution to the final protein concentration of 10 μM and final liposome concentration ranging from 2.5 mM to 0.625 mM. The JASCO J-1500 spectropolarimeter (JASCO, Tokyo, Japan) was used to register the CD spectra as previously described [11].

### 3.3. Differential Scanning Calorimetry Measurements

Calorimetric experiments were performed using the NanoDSC differential scanning calorimeter with 0.3 mL capillary cell (TA Instruments, New Castle, DE, USA) for the DPPA phospholipid dispersion, hCC V57G protein solution, and a mixed solution of DPPA and hCC V57G, at a scan rate of 1 °C/min. All measurements were performed in a filtered (0.22 µm) 20 mM PBS buffer at pH 7.4. Briefly, 1 mg of the DPPA phospholipid was dissolved in a 5 mL chloroform/methanol mixture (4/1, *v*/*v*). The organic solvent was then evaporated under a stream of nitrogen, and the formed lipid film was incubated under a vacuum to remove the residual organic solvent. The dry DPPA film was dispersed in PBS or a hCC V57G solution. The DPPA sample was then hydrated at 66 °C (10 °C above the lipid bilayer phase transition [48]) for 1 h with vigorous mixing. The sample containing DPPA and hCC V57G protein was vigorously mixed for 1 h at room temperature and sonicated for a few seconds in an ultrasonic bath to hydrate the lipid film. Both samples were stirred at room temperature for 1 h before measurement. The hCC V57G sample was prepared by dissolving the protein in PBS. The final concentration of DPPA and hCC V57G in the samples equaled 1.49 mM and 0.78 mM, respectively. All the samples were degassed for 10 min before scanning. Thermograms for the PBS buffer were subtracted from the excess heat capacity curves. The thermodynamic parameters were calculated and analyzed using Launch Nano Analyze software v. 4.0.1. (TA Instruments, USA). The lipid melting temperature (T_m_) and protein denaturation temperature (T_d_) were defined as the maxima of the temperature peaks, and the ΔT_1/2_ was defined as the width of the peak at half of its maximum height.

### 3.4. NMR Measurements

The NMR sample was formulated according to the previously described procedure [11]. The final concentrations of hCC V57G and DPPA equaled 0.15 mM and 0.01 mM, respectively. The NMR experiments were carried out as previously [11]. The NMRPipe [49] and NMRFAM-Sparky [50] software (build 04/22/21) was used for data processing and analysis, respectively.

### 3.5. ^15^N Relaxation Measurements

The ^15^N relaxation data were enquired with a ^1^H resonance frequency of 500.606 MHz and at the magnetic field of 11.7 T. The pulse sequences were adapted from the previously published data [46] and included in BioPack (Agilent, Santa Clara, CA, USA). The ^15^N *R*_1_ data sets and ^15^N *R*_2_ relaxation rates were obtained as described earlier [11]. The recycle delays and saturation times of ^1^H magnetization in both NOE experiments were kept at 6 s.

### 3.6. ^15^N Relaxation Data Analysis

The ^15^N *R*_1_ and *R*_2_ relaxation rates were calculated with the RELAX software (version 4.0.3) [51] for 82 amino acid residues. The ^15^N *R*_1_ and *R*_2_ values and the errors for relaxation parameters were calculated as described earlier [11]. The ^1^H-^15^N NOE values were calculated as saturated (I_sat_): reference (I_ref_) experiments ratio [52]. The obtained data were analyzed with the Spectral Density Mapping approach. The spectral density of molecular dynamics processes in the protein’s backbone amide groups was obtained at the frequencies of 0, ω_N_, and 0.87 ω_H_ [46].

### 3.7. Determination of Amino Acid Residues Buried in the DPPA Phospholipid Bilayer

The previously described method was applied to determine which amino acid residues were submerged into the DPPA bilayer [11]. Briefly, the chromium acetylacetonate (Cr(AcAc)_3_; Merck, Darmstadt, Germany) was mixed with the solution of ^15^N labeled hCC V57G and DPPA (C_hCC V75G_ = 0.15 mM, C_DPPA_ = 0.01 mM) at the final concentration of 50 mM. The results were estimated as the ^1^H-^15^N HSQC signal intensities ratio for a reference sample and after adding the Cr(AcAc)_3_ solution.

### 3.8. Molecular Dynamics Simulations

Coarse-grained molecular dynamics (CG MD) simulations were carried out using the MARTINI 3.0 force field [53] implemented in the GROMACS 2019.5 package [54]. CHARMM-GUI web-based graphical interface [55,56,57,58] was employed to build a fully hydrated lipid bilayer consisting of 400 DPPA molecules equally distributed between two leaflets. Two approaches were used to explore the interactions between the hCC V57G protein and the DPPA phospholipid bilayer. The first involved embedding the protein into the DPPA bilayer at the model-building stage using the CHARMM-GUI program. In the second approach, the hCC V57G protein was placed over a pre-equilibrated DPPA bilayer using the insert-molecules tool implemented in the GROMACS package (Appendix A). A coarse-grained model of hCC V57G protein was obtained from the NMR structure (PDB 6RPV [35]) using the martinize.py script (available on the Martini website on 5 February 2024). The tertiary structure of the CG hCC V57G protein during the CG MD simulations was maintained by an elastic network applied with a force constant of 500 kJ mol^−1^ nm^−2^ and a cutoff radius of 0.8 nm. The counterions were added to neutralize the net charge of the whole simulation system, and the ionic strength of the bulk solution was set to 150 mM NaCl. The systems were energy-minimized and equilibrated with the stepwise lowered force constant of the harmonic restraints (from 200 to 10 kJ mol^−1^ nm^−2^) to fix the position of the headgroups of the lipids in the DPPA bilayer. After equilibration, the systems were subjected to 5 µs isothermal-isobaric molecular dynamics (NPT) with a 20 fs time step. The target temperature was 338 K. The system was coupled to a temperature bath using the v-rescale temperature coupling scheme. The pressure was maintained semi-isotropically using the Parinello-Rahman barostat at 1 bar and a time constant of 12.0 ps. The van der Waals interactions were truncated at 1.2 nm. The long-range electrostatic interactions were treated using the Particle Mesh Ewald method and a cutoff of 1.2 nm. The dielectric constant was set to 15. Periodic boundary conditions were applied in all directions. The local membrane properties were analyzed using the g_lomepro software v. 1.0.2. [59]. The analysis of the remaining data, including lipid lateral diffusion coefficients, was performed using standard tools of the GROMACS package. Lipid lateral diffusion coefficients were determined from the linear-fitted slope of averaged two-dimensional mean square displacement (MSD) of the DPPA phosphate beads during the last 500 ns of CG MD simulations, while removing the effect of the center of mass motion of the bilayer and with resetting the initial reference point every nanosecond. Visualizations were performed in VMD 1.9.4a38 [60].

## 4. Conclusions

The study’s main objective was to determine whether the hCC protein exhibits affinity toward the DPPA bilayer. Generally, the results confirm the formation of the DPPA–hCC complex. As a result of the interaction, the protein structure is stabilized and increases its order. Based on the NMR data, it can be concluded that during the interaction, the hCC V57G protein is anchored to the membrane with the AS region, which is submerged into the DPPA bilayer. On the other hand, the MD simulations show that the protein can rearrange itself in space so that it may adopt various positions about the DPPA bilayer surface.

The results provide additional insight into the interactions between human cystatin C protein and liposomal membrane mimetics. It is consistent with the previous research, where the interactions between the hCC protein and DMPC [12] or POPG [11] phospholipid bilayers were monitored. In both cases, the protein is anchored to the bilayer via the AS region. The motives driving the protein–membrane interactions were similar, including Ser44, Asn82, and Asp119, and Ser44, Asp81, and Asp119 in the case of DMPC and POPG, respectively. When combined, these data indicate the AS region of the hCC protein and Ser44, Asn82, Asp81, and Asp119 amino acid residues as the general region of the protein responsible for interactions between the protein and cell membrane. Further, the increased organization and stabilization of the protein’s structure after the hCC–DPPA binding, which was not observed in the case of DMPC and POPG membrane mimetics, indicates the formation of a relatively stable protein–membrane complex. However, the data analysis does not allow for comparing the affinity level between the hCC protein and the three above-mentioned membrane mimetics. Such information would allow for determining the affinity between the hCC protein and different phospholipid membrane components, and potentially provide preliminary data for studies regarding the elucidation of more complex processes involving the hCC protein, e.g., transmembrane migration or membrane-mediated oligomerization. Therefore, further studies on the interactions between the hCC protein and biological membrane mimetics should focus on determining kinetic parameters for protein–membrane mimetic interactions. Additionally, further studies on interactions between hCC and membrane mimetics would benefit from increasing the complexity of the membrane mimetic composition to determine how the lipid content influences the affinity of the hCC protein toward the cell membrane.

## Figures and Tables

**Figure 1 molecules-29-03446-f001:**
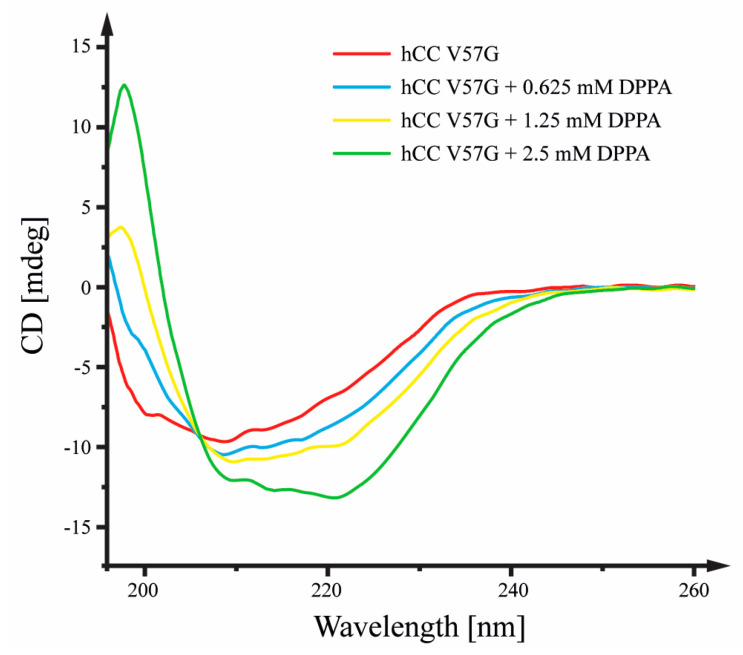
Circular dichroism spectra visualizing changes in the secondary structure of hCC V57G protein influenced by the environment of DPPA liposomes solution. Experiments performed at 30 °C. The concentration of hCC V57G equaled C_hCC V57G_ = 10 µM.

**Figure 2 molecules-29-03446-f002:**
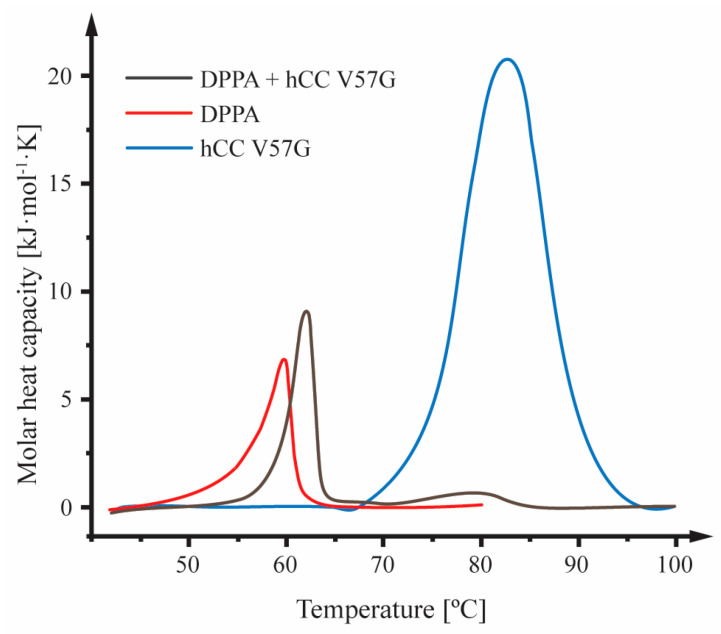
Molar heat capacity profiles for DPPA liposomes, hCC V57G protein, and a DPPA and hCC V57G mixture.

**Figure 3 molecules-29-03446-f003:**
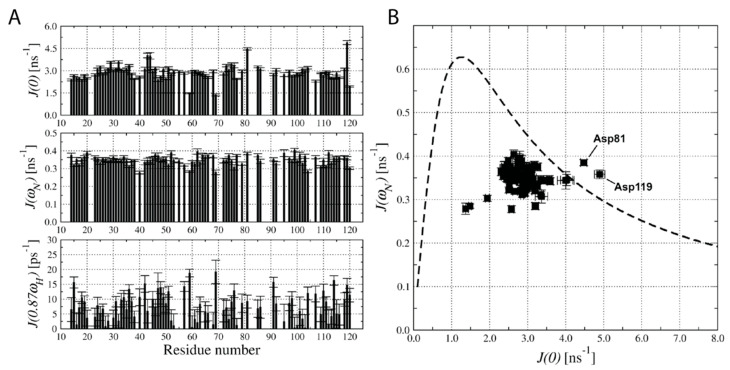
(**A**) Spectral density functions J(ω) calculated for frequencies 0, ω_N_, and 0.87 ω_H_, based on the ^15^N relaxation data acquired at 11.7 T and 298 K. (**B**) Graphical analysis of J(0) vs. J(ω_N_) spectral density values.

**Figure 4 molecules-29-03446-f004:**
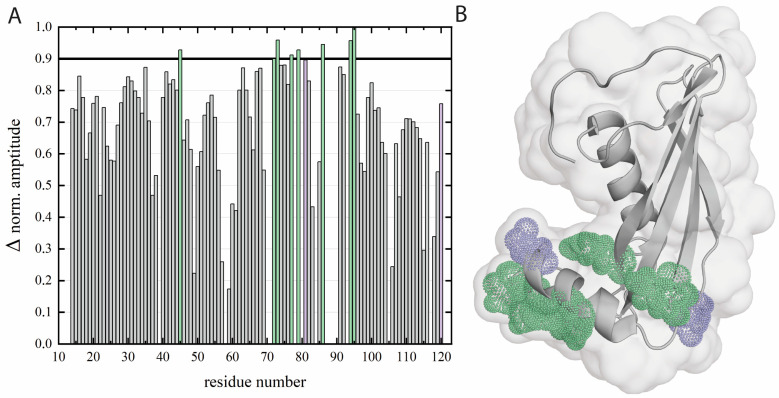
(**A**) Normalized amplitudes obtained based on the resonances in the ^1^H-^15^N HSQC spectrum registered for the hCC V57G protein in the presence of DPPA phospholipid bilayer, after the addition of the Cr(AcAc)_3_ paramagnetic compound. (**B**) The structure of hCC V57G protein (PDB: 6RPV), with amino acid residues exhibiting an amplitude change lower than 10% marked green. The residues with heightened molecular motion intensity in the low-frequency time frame are marked purple.

**Figure 5 molecules-29-03446-f005:**
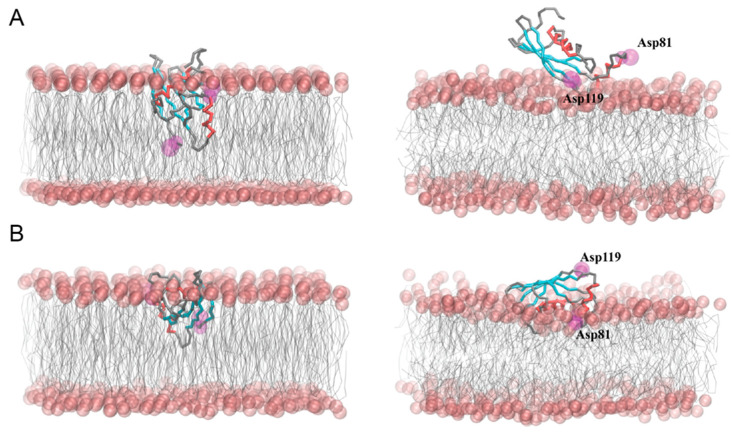
Initial (**left**) and final (**right**) snapshots from CG MD simulations of interactions between the hCC V57G and DPPA bilayer. Panels (**A**,**B**) correspond to models I and II, respectively, in which the protein was embedded in the DPPA phospholipid bilayer at the system-building stage. The phosphate groups of DPPA were shown as pink spheres. The β-sheet and α-helical protein fragments were marked in cyan and red, respectively. The Asp81 and Asp119 identified with NMR spectroscopy as potential driving factors in the protein–membrane interactions were marked in magenta in a space-filling model.

**Figure 6 molecules-29-03446-f006:**
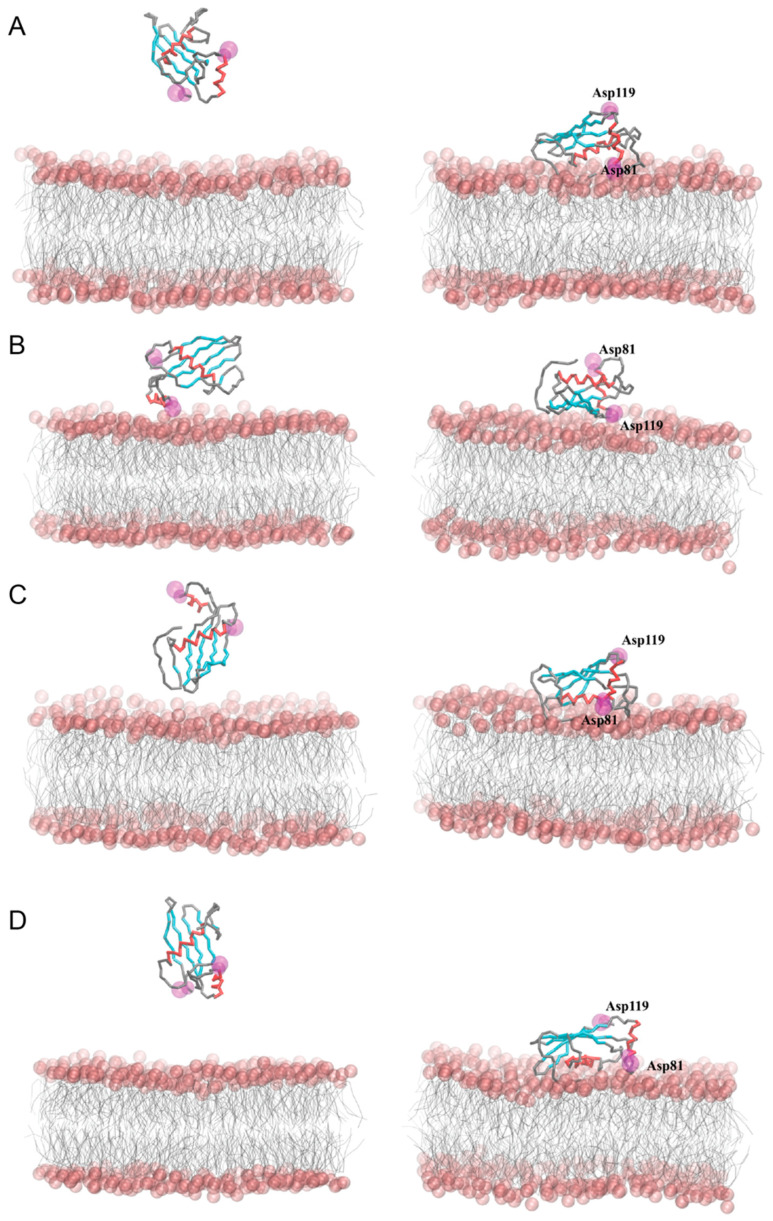
Initial (**left**) and final (**right**) snapshots from CG MD simulations of interactions between the hCC V57G and DPPA bilayer. Panels (**A**–**D**) correspond to models III–VI, respectively, with different initial distances and orientations of the protein in relation to the lipid bilayer. The phosphate groups of DPPA were shown as pink spheres. The β-Sheet and α-helical protein fragments were marked in cyan and red, respectively. The Asp81 and Asp119 identified with NMR spectroscopy as potential driving factors in the protein–membrane interactions were marked in magenta in a space-filling model.

**Figure 7 molecules-29-03446-f007:**
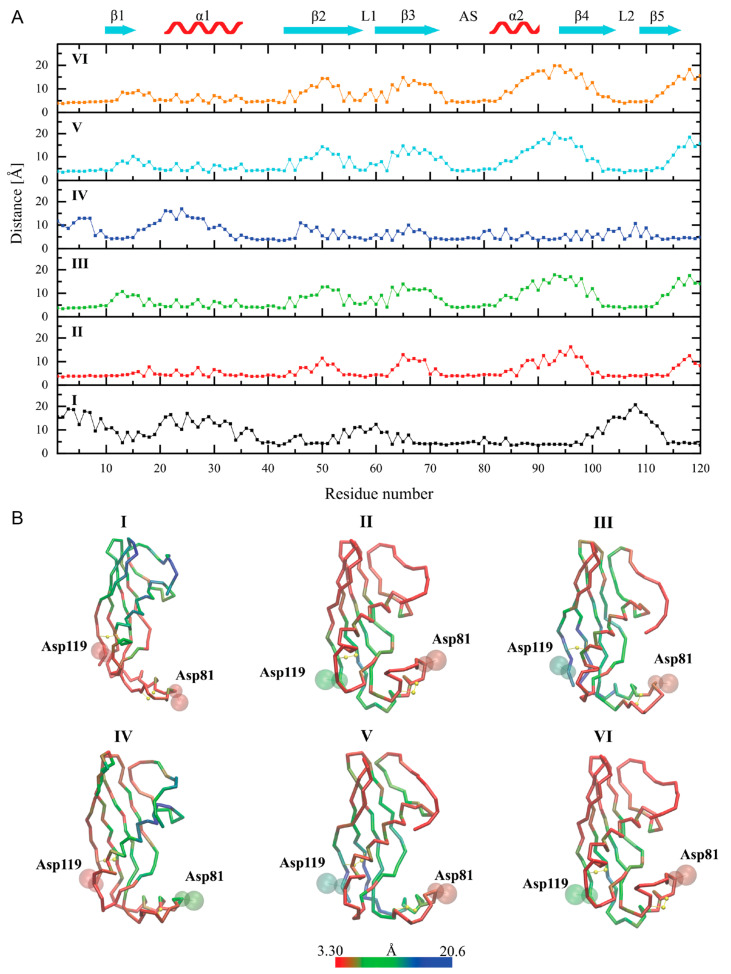
(**A**) Minimal distances between individual amino acid residues and DPPA lipids (distance between any pair of atoms) determined during the last 500 ns of CG MD simulations. (**B**) Final structures of the membrane-bound hCC V57G protein obtained from respective simulations (numbered). The colors correspond to the minimum distance between individual amino acid residues and DPPA lipids.

**Figure 8 molecules-29-03446-f008:**
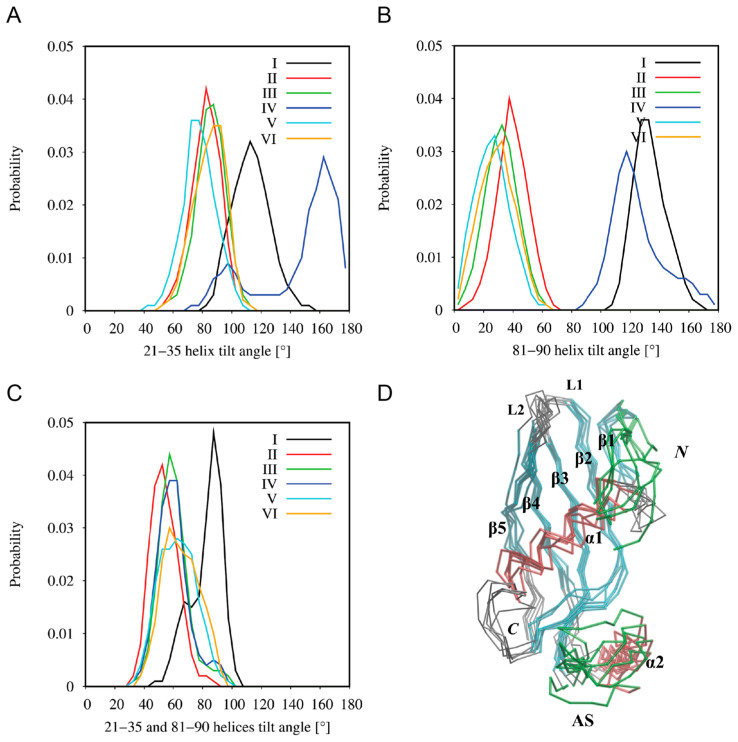
The distribution of tilt angles between (**A**) 21–35 α-helix (α1) and (**B**) 81–90 α-helix (α2) and the membrane normal, and (**C**) the distribution of tilt angles between 21–35 α-helix (α1) and 81–90 α-helix (α2) during the last 500 ns of the DPPA–hCC V57G binding CG MD simulations. The helix axes were defined as a vector connecting backbone beads of the respective helical N- and C-terminal residues. (**D**) Superimposed final structures of the membrane-bound hCC V57G protein obtained from the six respective CG MD simulations. Red color corresponds to helical fragment, cyan corresponds to β-strands, and green shows the most flexible parts of the protein.

**Table 1 molecules-29-03446-t001:** Thermodynamic parameters of the main phase transition of DPPA liposomes, hCC V57G protein, and a mixture of DPPA and hCC V57G in 20 mM PBS.

System	ΔH [kJ/mol]	ΔS [J/mol K]	T_m_ [°C]	T_d_ [°C]	ΔT_1/2_ [°C] ^a^
DPPA liposomes	32.8 ^b^	98.5 ^b^	59.8	-	3.50
DPPA liposomes	25.8 ^b^	76.9 ^b^	62.1	-	2.68
with hCC V57G	4.11 ^c^	11.6 ^c^	-	80.5	7.00
hCC V57G solution	281.9 ^c^	792.5 ^c^	-	82.6	10.8

^a^ the width at half-peak height; ^b^ determined for the melting peak of the DPPA phospholipid bilayer; ^c^ determined for the denaturation peak of the hCC V57G protein.

## Data Availability

The data presented in this manuscript is available on reasonable request from the corresponding author.

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
