# Peer review of "DPPA as a Potential Cell Membrane Component Responsible for Binding Amyloidogenic Protein Human Cystatin C"

_molecules, 2024, doi:10.3390/molecules29153446_

Round 1

Reviewer 1 Report

Comments and Suggestions for Authors

Igor Zhukov et al. extensively investigated the interaction between the model membrane DPPA and the amyloid-forming protein human cystatin C (hCC) using various experimental techniques and coarse-grained MD simulations. Overall, I find the experiments to be well designed and the results clearly presented. However, I have the following suggestions to improve the manuscript:

1.      The authors observed “an increase in the content of the α-helix in the protein structure” during interactions between hCC and DPPA liposomes using CD spectroscopy. Is this increase also observed in the CGMD simulations?

2.      In lines 122-124, the authors claim that “when compared to previous research on hCC-membrane interactions, this is the first time that phospholipid liposomes caused changes in the secondary structure of the hCC protein.” It is unclear whether this means that such changes are being observed for the first time or if it refers to the initial stage at which the protein undergoes such changes.

3. Both the experimental and computational findings are well presented. However, it is not clear to what extent the two approaches agree. Any behaviors observed in both approaches should be highlighted in the discussion to provide a clearer comparison and strengthen the conclusions.

Reviewer 2 Report

Comments and Suggestions for Authors

The manuscript entitled “DPPA as a potential cell membrane component responsible for binding amyloidogenic protein human cystatin C shown that the interactions between a model membrane (DPPA) and one of amyloid forming proteins-human cystatin C. However, the manuscript needs minor revision before it can be accepted in this journal, as explained below:

Comments 1:It is suggested that fonts be unified in manuscripts.

Comments 2: It is suggested that an abbreviation list can be added.

Comments 3: Whether the results are statistically significant?min

Comments on the Quality of English Language

minor editing.

Reviewer 3 Report

Comments and Suggestions for Authors

The authors present a new study regarding the interaction between the hCC V57G protein and DPPA phospholipid bilayers, suggesting that the hCC V57G protein stabilizes the lipid bilayer, increasing its uniformity and coherence. They also propose that the AS region of hCC dips into the DPPA bilayer, inducing structural changes in the protein.

There are several points that need to be addressed to improve the reliability and reproducibility, as follows.

1. Quantitative analysis of the binding affinity and stoichiometry of the hCC V57G-DPPA interactions, if possible -DMPC and -POPG interactions [1,2], using techniques such as surface plasmon resonance (SPR) or isothermal titration calorimetry (ITC). These data would support the qualitative date of the NMR, CD and MD simulations in this study.

2. Since the protein oligomerization in amyloid diseases is important [3,4], the authors should show the effect of DPPA on the oligomeric state of hCC V57G. For example, experiments using size exclusion chromatography (SEC) or native gel electrophoresis could examine whether the interaction with DPPA would promote or inhibit the formation of hCC V57G oligomers.

3. The authors should consider the physiological concentration ranges of DPPA, if possible including DMPC and POPG, and hCC in different tissues and cell types, to discuss the biological relevance and importance of their findings. The authors should discuss how the observed interactions may contribute to the function or dysfunction of hCC in vivo and the potential role in the pathogenesis of amyloid diseases.

References:

[1] Jurczak, P., et al. (2020). DMPC Phospholipid Bilayer as a Potential Interface for Human Cystatin C Oligomerization: Analysis of Protein-Liposome Interactions Using NMR Spectroscopy. Membranes, 11(1), 13.

[2] Jurczak, P., et al. (2024). Monitoring the interactions between POPG phospholipid bilayer and amyloid-forming protein human cystatin C. Does the bilayer influence the oligomeric state and structure of the protein? Biochimica et Biophysica Acta (BBA) - Biomembranes, 1866(3), 184285.

[3] Butterfield, S. M., & Lashuel, H. A. (2010). Amyloidogenic protein-membrane interactions: mechanistic insight from model systems. Angewandte Chemie International Edition, 49(33), 5628-5654.

[4] Sciacca, M. F., et al. (2018). Amyloid growth and membrane damage: Current themes and emerging perspectives from theory and experiments on Aβ and hIAPP. Biochimica et Biophysica Acta (BBA) - Biomembranes, 1860(9), 1625-1638.

Comments on the Quality of English Language

Minor editing of English language required.

Reviewer 4 Report

Comments and Suggestions for Authors

1) For "The interactions 111 with the DPPA caused an increase of strength of the signals in CD spectra at wavelengths 112 of ca. 209 nm and 221 nm (Fig. 1), indicating an increase of content of the α-helix in the 113 protein structure. The formation of an α-helix and an increase of order within the protein 114 structure was most probably induced by the interaction of the AS region (marked in the 115 Fig. S2, Supporting Information), a relatively unorganized fragment of the protein, with 116 the hydrophobic environment of the DPPA bilayer interior." Kindly comment about the density of solvent in this region after formation of α-helix.

2)  Explain for this "Similarly to our previous studies [11], the rotational diffusion tensor can be described as axially symmetrical model of the protein." why rotational tensor is again successfully described by symmetrical model of the protein. Did authors explore some other possibility?

3) Does  any probable errors in1H-15N NOE have been checked by any sound criterion of nonlinear regression methods.

4) Mention the ensemble employed for all MD simulations.

5) Provide some citations in MD section introduction after "The coarse-grained molecular dynamics simulations (CG MD)". like:

Pederzoli, Marek, Mirza Wasif Baig, Mojmir Kyvala, Jiri Pittner, and Lukasz Cwiklik. "Photophysics of BODIPY-based photosensitizer for photodynamic therapy: Surface hopping and classical molecular dynamics." Journal of Chemical Theory and Computation 15, no. 9 (2019): 5046-5057.

Round 2

Reviewer 3 Report

Comments and Suggestions for Authors

Regarding Comment 1:

> Results of those comparison studies will be described in another manuscript, which is currently under review.

If these data have already been submitted to another journal, it is reasonable not to include them in the current report. However, given the authors' previous studies, it seems somewhat odd that the DMPC experiments were not performed in parallel. Furthermore, it is assumed that the kon and koff data should be presented together with the KD values.

Regarding Comment 2:

Based on the response, the hCC V57G mutant, which is easier to handle as a monomer molecule, was used in this paper to obtain the results. Therefore, the results of this paper are based on this easy to handle molecule. However, it is crucial to confirm by additional experiments that the hCC V57G mutant behaves similarly to the wild type (WT) in order to understand the original function of human cystatin C. Since this important point is missing throughout the text, it is necessary to add an additional experiment or at least address this point in the Discussion section.

Regarding Comment 3:

> Further, our studies (including unpublished data) show that neither of the lipids (DPPA, DMPC, POPG) causes the oligomerization of the hCC protein.

This answer is interesting because it contradicts the answer given in comment 2. It would be helpful to specify the experimental methods used to verify the oligomerization of hCC. In addition, if the authors believe that lipids do not affect oligomerization, the following sentence in the Introduction is misleading. It would be more appropriate to state that lipids modulate oligomerization to some extent:

phospholipid membrane is also involved in pathological processes leading to harm and disease by playing an important role in the mechanisms ““leading”” to the oligomerization of amyloid forming proteins

Comments on the Quality of English Language

As mentioned in the initial review, there are still some minor grammatical errors throughout the text, so it is better or necessary to have the entire text proofread by a native speaker.
